# Image-based metric of invasiveness predicts response to adjuvant temozolomide for primary glioblastoma

Susan Christine Massey[1]*, Haylye White[1], Paula Whitmire[1], Tatum Doyle[1,2], Sandra K. Johnston[1,3], Kyle W. Singleton[1], Pamela R. Jackson[1], Andrea Hawkins-Daarud[1], Bernard R. Bendok[4,5,6,7], Alyx B. Porter[8], Sujay Vora[9], Jann N. Sarkaria[10], Leland S. Hu[5], Maciej M. Mrugala[8], Kristin R. Swanson[1,4,11]

1 Mathematical NeuroOncology Laboratory, Precision Neurotherapeutics Innovation Program, Mayo Clinic, Phoenix, Arizona, United States of America, 2 College of Literature, Science and the Arts, University of Michigan, Ann Arbor, Michigan, United States of America, 3 Department of Radiology, University of Washington, Seattle, Washington, United States of America, 4 Department of Neurologic Surgery, Mayo Clinic, Phoenix, Arizona, United States of America, 5 Department of Radiology, Mayo Clinic, Phoenix, Arizona, United States of America, 6 Department of Otorhinolaryngology (ENT)/Head and Neck Surgery, Mayo Clinic, Phoenix, Arizona, United States of America, 7 Neurosurgery Simulation and Innovation Laboratory, Precision Neurotherapeutics Innovation Program, Mayo Clinic, Phoenix, Arizona, United States of America, 8 Department of Neurology, Mayo Clinic, Phoenix, Arizona, United States of America, 9 Department of Radiation Oncology, Mayo Clinic, Phoenix, Arizona, United States of America, 10 Department of Radiation Oncology, Mayo Clinic, Rochester, Minnesota, United States of America, 11 School of Mathematical and Statistical Sciences, Arizona State University, Tempe, Arizona, United States of America

☯ These authors contributed equally to this work.
* Massey.Susan@mayo.edu

**Data Availability Statement:** The de-identified minimal data set used for our analysis has been provided as supplemental material. The raw imaging data, which requires additional human

## Abstract

### Background

Temozolomide (TMZ) has been the standard-of-care chemotherapy for glioblastoma (GBM) patients for more than a decade. Despite this long time in use, significant questions remain regarding how best to optimize TMZ therapy for individual patients. Understanding the relationship between TMZ response and factors such as number of adjuvant TMZ cycles, patient age, patient sex, and image–based tumor features, might help predict which GBM patients would benefit most from TMZ, particularly for those whose tumors lack $O^6$–methyl-guanine–DNA methyltransferase (MGMT) promoter methylation.

### Methods and findings

Using a cohort of 90 newly–diagnosed GBM patients treated according to the standard of care, we examined the relationships between several patient and tumor characteristics and volumetric and survival outcomes during adjuvant chemotherapy. Volumetric changes in MR imaging abnormalities during adjuvant therapy were used to assess TMZ response. T1Gd volumetric response is associated with younger patient age, increased number of TMZ cycles, longer time to nadir volume, and decreased tumor invasiveness. Moreover, increased adjuvant TMZ cycles corresponded with improved volumetric response only

subjects protections, is available upon request to Mayo Clinic Legal Contract Administration, Phone: +1 (507) 266-1326.

**Funding:** This work was supported by NIH R01NS060752 (KRS), R01CA164371 (KRS), U54CA210180 (JNS,KRS), U54CA143970 (KRS), U54CA193489 (KRS), and U01CA220378 (KRS), the Arizona Biomedical Research Commission ADHS16-162514 (KRS), the James S. McDonnell Foundation, and the Ben & Catherine Ivy Foundation. The funders had no role in study design, data collection and analysis, decision to publish, or preparation of the manuscript.

**Competing interests:** The authors have declared that no competing interests exist.

among more nodular tumors, and this volumetric response was associated with improved survival outcomes. Finally, in a subcohort of patients with known MGMT methylation status, methylated tumors were more diffusely invasive than unmethylated tumors, suggesting the improved response in nodular tumors is not driven by a preponderance of MGMT methylated tumors.

## Conclusions

Our finding that less diffusely invasive tumors are associated with greater volumetric response to TMZ suggests patients with these tumors may benefit from additional adjuvant TMZ cycles, even for those without MGMT methylation.

## Introduction

Glioblastoma (GBM) is the most common malignant primary brain tumor found in adults [1,2]. Despite diligent research efforts, patients diagnosed with this aggressive cancer have a one–year, two-year, and five-year average survival rates of 40.8%, 18.5%, and 6.8%, respectively [2]In 2005, Stupp et al. found that maximal safe resection followed by concurrent radiotherapy and temozolomide (TMZ) chemotherapy and six adjuvant cycles of TMZ resulted in a median overall survival of 14.6 months compared to 12.1 months for radiotherapy alone. Used as the control arm in the EF-14 trial evaluating tumor treating fields, the median overall survival for patients receiving this protocol was 16 months [3]. Today, this protocol remains the standard-of-care for patients diagnosed with GBM.

As an alkylating agent, TMZ operates by methylating and damaging DNA, preventing proliferation and inducing apoptosis [4]. Compared to other therapeutic agents, TMZ is relatively blood-brain barrier (BBB) penetrant, with a CSF to plasma ratio of 33% [5], which is one factor that makes it effective against gliomas. The angiogenic nature of glioblastoma causes the breakdown of the BBB in the vicinity of the tumor, which also contributes to the drug's ability to reach the tumor cells [6,7]. In addition to inducing the apoptosis of glioma cells, TMZ in combination with radiotherapy can cause pseudoprogression, which is observed as progressive imaging changes that look similar to true progression and are thought to be a result of treatment-induced inflammation [8]. The similar radiological presentation of growing tumor and pseudoprogression complicates the assessment of TMZ response [9]. Some reports suggest that waiting until after three cycles of adjuvant TMZ (i.e., approximately 12 weeks from completion of radiotherapy) to assess treatment response can improve the accuracy of progression determination [10].

Patients typically receive a daily TMZ dose of 75 mg per square meter of body-surface area during radiotherapy, followed by a dose of 150–200 mg per square meter for 5 days during each 28–day adjuvant cycle for 6–12 cycles. TMZ is generally well tolerated, with about one-third of patients experiencing nausea and vomiting that is typically well controlled by antiemetics [11]. Patients are also at risk for infection, lymphopenia, neurotoxicity [12], or hematologic toxicities, such as thrombocytopenia [9]. Stupp et al. found that in a population of over 200 patients, the percentage of people who discontinued therapy due to the toxic effects of TMZ was only 5% during the concurrent stage and 8% of patients during the adjuvant stage [13]. The FDA labeling specifies giving six cycles of adjuvant TMZ, although the number of cycles of adjuvant TMZ administered in clinical practice varies. Administration of the drug may be discontinued early due to adverse effects or disease progression, while some patients

and their physicians elect to administer the drug beyond 6–12 cycles, sometimes for as long as 2–3 years or until ultimate disease progression [14]. The relationship between number of adjuvant cycles received and outcome has not been clearly elucidated. Three studies found that patients who received more than 6 cycles of TMZ had improved survival compared to those who received less [15–17], while two other studies found no survival difference between these two groups of patients [18,19]. Taken together, these suggest that an as yet unidentified subset of patients may derive benefit from more than 6 cycles of TMZ and drive this difference when observed.

Even when patients receive the same number of cycles of TMZ without adverse effect, there can be large variation in tumor response. This is largely attributed to particular molecular features (genetic and epigenetic), which may predispose a patient to a better TMZ response and/ or delayed evolution of TMZ resistance. The molecular feature that is given the most attention in regards to TMZ sensitivity is O6-methylguanine-DNA methyltransferase (MGMT) promoter methylation [9]. Methylation of the MGMT promoter in GBM effectively silences this DNA repair gene, making tumor cells unable to repair the cytotoxic O6-methylguanine lesions induced by TMZ and some other alkylating agents [20–22]. MGMT promoter methylation exists in about 35% of GBMs [23] and is associated with longer overall and progression-free survival [24,25]. Research has suggested that tumor responsiveness to TMZ is also impacted by IDH1 mutation and p53 mutation in lower grade gliomas [26,27].

While genetic differences are currently the best supported predictors of TMZ response, recent studies have found that other patient characteristics impact TMZ response. Recently, a large-scale investigation found that female GBM patients live longer than male GBM patients [28]. Considering that TMZ is a part of standard-of-care practice, this raises the question of whether there is an impactful sex difference in tumor responsiveness to TMZ. It has been observed that females have an improved volumetric response and exhibit better tumor control during adjuvant TMZ than males [29,30], but further research is needed to fully elucidate the biological mechanism of this sex difference. Age is recognized as a significant prognostic indicator for GBM patients. It is further thought that older patients are not as tolerant to aggressive treatment as their younger counterparts [31], and patients older than 70 years were not included in the study that established the current standard-of-care [13]. While toxicity and adverse reactions remain a concern, a prospective study on GBM patients 65 years or older found that adding adjuvant TMZ to a radiotherapy treatment course improves median overall survival by 3.7 months and PFS by 5.4 months [32], and later studies examining alternative radiotherapeutic regimens have also found TMZ to improve survival in older patients [33,34]. This substantial impact on outcome emphasizes the need to assess whether age impacts tumor responsiveness to TMZ.

Considering the large variation in response to TMZ, the potential for adverse reaction, and the uncertainty caused by pseudoprogression, deciding how many cycles of adjuvant TMZ to administer to a patient is a challenging task for clinicians. The possibilities of adding another therapy during adjuvant TMZ, such as tumor treating fields (TTF), or continuing administration of TMZ beyond six cycles adds further complexity to clinical decision making. Outside of the presence of MGMT methylation, which does not apply to a majority of patients, there are few clear indicators to aid clinicians in this process. In this investigation, we sought to identify image-based characteristics associated with TMZ response that can be assessed in the pre-adjuvant setting. By comparing pre-adjuvant and post-adjuvant MR images, we sought characteristics that are associated with volumetric response, overall survival, and progression free survival. Additionally, we examined how the number of TMZ cycles received and MGMT methylation status influenced these relationships.

## Methods

### Patient cohort

Our lab has amassed a multi-institutional repository of over 1400 glioma patients, which has been approved by the Mayo Clinic Institutional Review Board (IRB) studies 17–009682, 17–009688, and 15–002337. Data were acquired from a variety of sources, including anonymized data from collaborating institutions and publicly available data. For patients that enrolled we obtained either: 1) written informed consent from patients after given ample opportunity to review the consent form and ask questions before deciding to sign consent, or 2) waiver of patient consent was granted from the IRB for retrospective studies and for special circumstances (e.g. in the event an eligible person was identified for the study but was discovered to be deceased). The cohort for this present study consists of all subjects in this repository who met the following criteria: A) diagnosed with primary GBM (n = 1323), B) received maximal safe resection, concurrent radiation therapy (XRT) and TMZ, and at least one adjuvant cycle of TMZ (n = 234), C) had available age at diagnosis, sex, overall survival, and treatment start/stop dates (n = 210), D) did not receive any therapies other than XRT, TMZ, anti-seizure medications, or steroids between the first surgery and first cycle of adjuvant TMZ (n = 175), and E) had sufficient pre-adjuvant and post-adjuvant MR imaging (detailed below in "Imaging and Biomathematical Model") (n = 90). These inclusion criteria resulted in the identification of a cohort of 90 patients (Table 1). Eleven patients received a therapy other than TMZ concurrent with or in between cycles of TMZ; these other therapies included additional resection or radiotherapy, thalidomide, accutane, and bevacizumab. While the concurrent use of TMZ and TTF is becoming more common, none of the patients in this cohort received TTF during adjuvant therapy. Further, to ensure that we captured the effect of TMZ exclusively in these eleven cases, the image before the start of the other therapy was used as the post-adjuvant image, so that no patients received other therapies during the analyzed imaging period.

### Imaging and biomathematical model

We defined "adjuvant TMZ" as the time period when patients consistently received cycles of TMZ alone after the completion of surgery and concurrent TMZ and XRT. If patients received another therapy concurrent with TMZ or between cycles of TMZ, we only considered the period when they received TMZ alone to be "adjuvant TMZ" and excluded the cycles administered after the start of the other therapy. Further, in order for patients to be included in this study, they had to have the following available MR images: 1) gadolinium enhanced T1-weighted (T1Gd) and T2-weighted or T2 fluid-attenuated inversion recovery (T2-FLAIR) images between concurrent XRT/TMZ and adjuvant TMZ ("pre-adjuvant" images), and 2) T1Gd and T2-FLAIR images near the dates of their last cycle of adjuvant TMZ (taken either 1–2 cycles before the end of adjuvant therapy or up to 40 days after) ("post-adjuvant" images). All post-adjuvant images were taken after the administration of cycles of TMZ only and before the start of any other therapy. Tumor volumes were segmented as regions of imaging hyperintensity on T1Gd and T2/FLAIR MRI sequences using an in-house, python-based, semi–automated segmentation software leveraging intensity thresholding. Each image was segmented by at least one trained technician and reviewed by an expert for quality assurance and consistency. On occasion, tumor volumes were segmented by more than one technician; in these cases, the measured volumes were averaged. These tumor volumes were then converted to spherically equivalent tumor radii for use in this investigation (T1Gd radius and T2-FLAIR radius). From the T1Gd radii we computed the percent change of the T1Gd radius over the adjuvant TMZ cycles (%Δ T1Gd), which is

**Table 1. Distributions and counts of relevant demographic, volumetric, and treatment-based patient characteristics.**

| | N = | Mean | Median | Range |
|---|---|---|---|---|
| **Sex** | | | | |
| Male | 60 (66.7%) | ----- | ----- | ----- |
| Female | 30 (33.3%) | ----- | ----- | ----- |
| **Age (years)** | 90 | 54.66 | 57.5 | 18–76 |
| **Overall Survival (days)** | | | | |
| Confirmed death | 71 (78.9%) | 806.0 | 562 | 115–3245 |
| Alive/Lost to follow-up | 19 (21.1%) | 1404 | 1278 | 128–3819 |
| **Time from adjuvant TMZ to progression (days)[b]** | 43 (47.8%) | 241.0 | 30 | 7–1709 |
| **Extent of Resection** | | | | |
| Gross Total Resection | 40 (44.4%) | ----- | ----- | ----- |
| Sub-total Resection | 35 (38.9%) | ----- | ----- | ----- |
| Biopsy | 15 (16.7%) | ----- | ----- | ----- |
| **Cycles of adjuvant TMZ[a]** | 90 | 6.122 | 5 | 1–21 |
| Received <6 cycles | 47 (52.2%) | ----- | ----- | ----- |
| Received 6 cycles | 14 (15.6%) | ----- | ----- | ----- |
| Received 7+ cycles | 29 (32.2%) | ----- | ----- | ----- |
| **Pre-adjuvant D/rho (mm$^2$)** | 90 | 2.073 | 1.409 | 0.0034–9.525 |
| **Pre-adjuvant T1Gd radius (mm)** | 90 | 12.03 | 10.81 | 2.312–32.25 |
| **Post-adjuvant T1Gd radius (mm)** | 90 | 11.90 | 11.62 | 0.00–22.22 |
| **%Δ T1Gd** | 90 | 7.70% | -0.16% | -100%–260% |

Extent of resection is abstracted from surgical notes and radiological reports and is not uniformly verified radiographically (though most GTR cases were verified via imaging). Distributions of the nadir-related variables are in S1 Table.

[a]The cycles of adjuvant TMZ reported here exclude any cycles that were given in conjunction with other anti-tumor therapies since these were excluded from our analysis (see Methods). It should be noted that the majority of patients did receive at least 6 cycles of TMZ, even if they were not counted for the adjuvant period in our analysis.

[b]These are results for subjects with a known date of true progression only, which was slightly less than half of the total cohort (47.8%).

calculated by finding the difference between the post-adjuvant T1Gd radius and pre-adjuvant T1Gd radius and dividing it by the pre-adjuvant T1Gd radius.

Next, using the pre-adjuvant T1Gd and T2-FLAIR images, we calculated a mathematical model–based tumor invasion metric called D/rho. This metric derives from the proliferation-invasion (PI) model of glioblastoma growth [35,36] and describes the ratio of overall tumor invasion to proliferation, with higher D/rho indicating a more diffuse tumor and lower D/rho indicating a more nodular tumor [37,38]. Diffuse tumors have higher levels of model-predicted net cellular invasion relative to model-predicted net cellular proliferation, while nodular tumors have more proliferation relative to invasion into the surrounding tissues. We computed this metric at the pre-adjuvant imaging time point to establish a baseline of tumor invasiveness prior to the administration of adjuvant TMZ.

## Response indicator

For the purposes of this investigation, we split our cohort into "responders" and "non-responders" based on the tumor volume changes observed in T1Gd images over the course of adjuvant therapy. Patients that had a decrease in T1Gd abnormality volume (negative %Δ T1Gd) following adjuvant TMZ were considered "responders" and patients that had an increase in volume (positive %Δ T1Gd) were considered "non-responders". Note that this classification is not intended for clinical decision-making or to distinguish progression from stable

disease; therefore, in this investigation, these terms refer exclusively to T1Gd volumetric response and "outcome" refers to survival. While we compared pre-adjuvant T1Gd volume with post-adjuvant T1Gd volume for the calculation of %Δ T1Gd, we also conducted an investigation that compared the pre-adjuvant volume with the nadir volume (%ΔT1Gd-Nadir). This nadir volume is defined as the smallest T1Gd volume at any time point after the pre-adjuvant time point and before or at (if a smaller volume was not reached earlier or no intermediate images were available) the post-adjuvant time point. Note that pseudoprogression, when it occurs, can increase our imaging-based measure of tumor volume. This could result in some tumors being misclassified as responders due to the resolution of pseudoprogression during the course of adjuvant TMZ. While this remains a potential confounder for many GBM studies, our classification correlated well with overall and progression free survival, suggesting that any such misclassification was minimal. To further reduce the possibility of misclassification due to pseudoprogression, we re-performed all of our analyses in a supplemental investigation using a subcohort of patients with more than 12 weeks between the end date of XRT and date of post-adjuvant imaging (n = 72) (S7–S10 Figs).

## Statistical analysis

Two-sided t-tests with Welch's corrections were used to test for differences in the means of two groups. F-tests with linear regression models were used to test whether two variables had a significantly positive or negative correlative relationship or were not related. Kaplan-Meier curves were used to visualize survival data and log-rank tests were used to test whether two groups had significantly different outcomes. All of these statistical tests were performed using R [39,40] using packages *survival* [41], *survminer* [42], and *ggplot2* [43]. A p-value of 0.05 was used as the cut-off for statistical significance.

## Study approval

All patients included in this investigation were consented prospectively or approved for retrospective research by institutional review boards.

## Results

### T1Gd volumetric response correlates with younger patient age, increased number of TMZ cycles, longer time to nadir, and decreased tumor invasiveness

In order to understand whether various patient or tumor characteristics were significant predictors of tumor response to TMZ, we classified patients as "responders" or "non-responders" based on change in T1Gd volume over the course of adjuvant TMZ, as detailed in the Methods. Responders (n = 45) were younger (t-test, p = 0.0450), received more cycles of TMZ (p<0.0001), reached nadir later during the adjuvant time period (p = 0.0046), and had tumors that were more nodular (p = 0.0191) than non-responders (n = 45) (Fig 1). MR images of a nodular responding patient and a diffuse non-responding patient are shown as examples in Fig 2. There was no difference in the pre-adjuvant tumor volume (T1Gd radius p = 0.1007, T2-FLAIR radius p = 0.719) between responders and non-responders. Responders also had significantly longer survival than non-responders (log-rank test, p = 0.0028) (Fig 3). Among patients with a recorded date of progression, responders (n = 21) tended to have a longer time between TMZ and progression than non-responders (n = 22) (p = 0.0674) (Fig 3). The relationship between the T1Gd-based volumetric response and outcome validates its relevance as an indicator of treatment response. Meanwhile, changes in T2-FLAIR abnormality had no

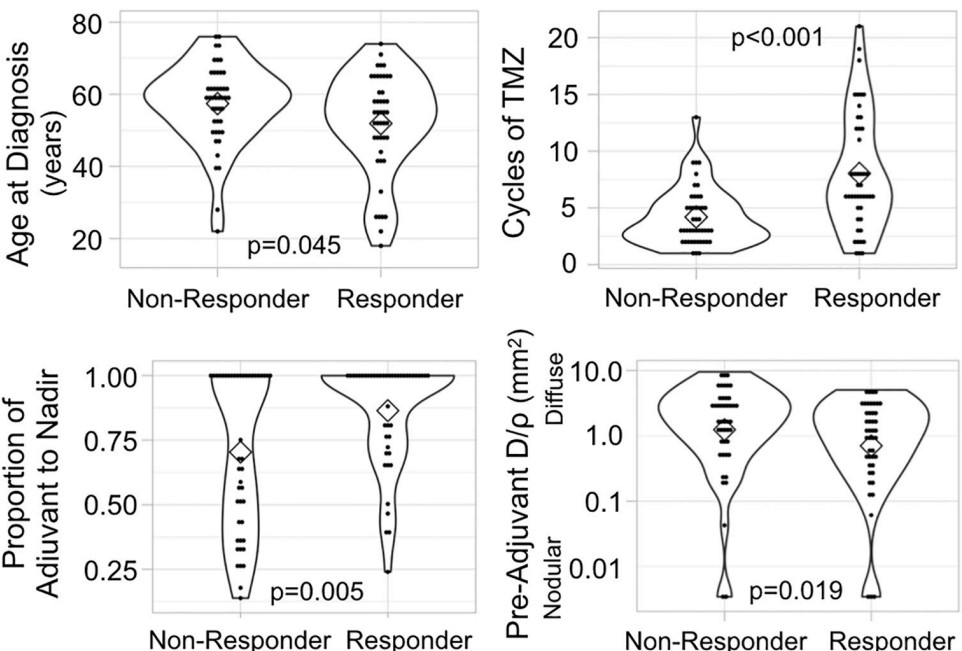

**Fig 1. Characteristic differences between responders (n = 45) and non-responders (n = 45).** Statistical tests (t-tests) show that volumetric responders (decrease in T1Gd volume during adjuvant TMZ) were younger, received more cycles of TMZ, reached nadir relatively later during adjuvant therapy, and had more nodular tumors than non-responders (increase in T1Gd volume during adjuvant TMZ). Proportion of adjuvant to nadir is calculated as the number of days between pre-adjuvant and nadir images divided by the total number of days between pre-adjuvant and post-adjuvant images. (Black dots in the violin plots indicate individual subject values, diamonds indicate median values, and outline denotes frequency).

clear relationship with patient outcome. Therefore, quantifying the changes in T2-FLAIR radius does not appear to provide a valuable response indicator.

## Increasing number of cycles correlates with volumetric response only in nodular tumors

In order to test the impact of tumor invasiveness on volumetric response and outcome, we divided the patients into three equally sized groups based on pre-adjuvant D/rho (nodular, moderate, and diffuse). When we considered the impact of number of cycles on volumetric response, we found a significant negative correlation between cycles of TMZ received by a patient and their %Δ T1Gd among nodular tumors (F-test, p = 0.0062), but this relationship did not exist among diffuse tumors (p = 0.4040) (Fig 4). This indicates that additional cycles of TMZ have a clearer volume reduction benefit among patients with nodular tumors than among those with diffuse ones. This volumetric benefit is also tied to outcome, with nodular tumors having a distinct relationship between volumetric change and survival. Specifically, we found that the survival difference observed between responders and non-responders is only significant among the nodular tumors (log-rank, nodular p = 0.0021, diffuse p = 0.793) (Fig 4).

## MGMT methylated tumors are more diffusely invasive

Since methylation of the MGMT promoter corresponds with improved TMZ response, we investigated whether our findings might simply be attributable to a co-occurrence of those features with MGMT methylation. Using our limited sample of patients with known MGMT

## Nodular Responder
### 68 y.o. male, 15 TMZ cycles
### Pre-Adjuvant D/ρ = 0.207

## Diffuse Non-Responder
### 59 y.o. male, 6 TMZ cycles
### Pre-Adjuvant D/ρ = 7.495

**Fig 2. Pre-adjuvant and post-adjuvant T1Gd and T2-FLAIR MR images of a nodular responding patient and a diffuse non-responding patient.** The spherically-equivalent radius converted from the volume of each lesion is listed below the image in millimeters; these were used to derive the D/rho diffusivity index. (y.o. = years old).

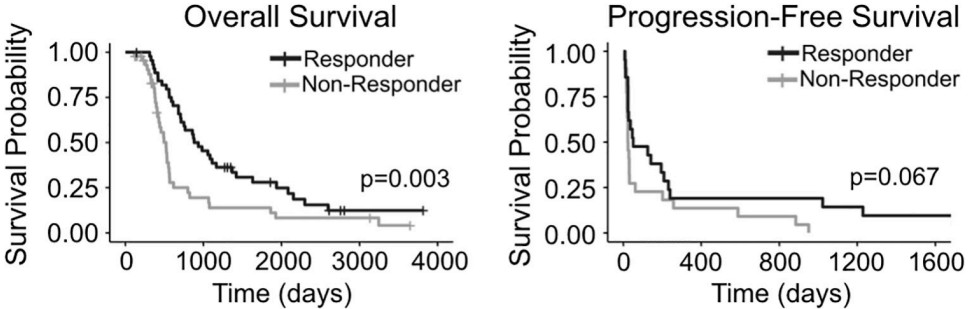

**Fig 3. Survival comparison between responders and non-responders.** Responders (n = 45, decrease in T1Gd volume during adjuvant TMZ) had significantly longer overall survival than non-responders (n = 45, increase in T1Gd volume during adjuvant TMZ). Among patients with dates of progression, responders (n = 21) tended to have longer times to progression than non-responders (n = 22).

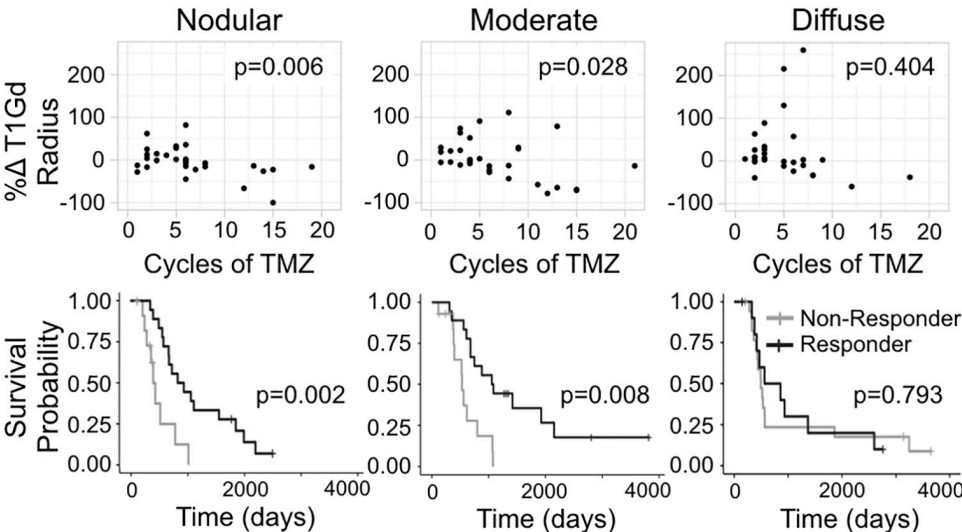

**Fig 4. Percent change T1Gd radius vs cycles of TMZ and survival probabilities for responders and non-responders grouped by nodular, moderate, and diffuse tumors. Subjects** were split into three evenly sized groups based on their pre-adjuvant D/rho values: lowest third "nodular" (0.0034 to 0.572 mm2), "moderate" (0.6195 to 2.562 mm2), and the highest third "diffuse" (2.567 to 9.53 mm2). Among the nodular tumors (n = 30), there is a significant negative correlation between volumetric response and cycles of TMZ received. Then this improved response is clearly tied to outcome since nodular responders (based on T1Gd volume change) had significantly longer survival than non-responders of the same group. The relationships between cycles, response, and outcome are not significant among diffuse tumors (n = 30).

methylation status (methylated n = 9, unmethylated n = 14), we analyzed the relationship between methylation status, tumor volumetric response, cycles of TMZ, and D/rho (S4 and S5 Figs). Patients with MGMT methylated tumors had significantly better survival than those with unmethylated tumors (log-rank, p = 0.014), consistent with existing literature. Further, those with MGMT methylated tumors are more commonly responders (6 responders vs. 3 non-responders) and have significantly better volumetric response than those with unmethylated tumors (t-test, p = 0.024). Among the volumetric responders (n = 11), MGMT methylation (n = 6) showed a survival benefit over unmethylation (n = 5) (p = 0.014). However, patients with methylated tumors also received more cycles of TMZ (p = 0.0156), which could indicate that prescribing practices have created a confounding factor in the relationship between methylation and volumetric response. Focusing within methylated tumors, we observe a clear negative correlation between cycles of TMZ received and %Δ T1Gd during adjuvant TMZ (Fig 5), similar to that among nodular tumors. This comparison is limited by its small sample size which lacks statistical power, but the observation supports the existing idea that methylated tumors respond well to TMZ chemotherapy. Interestingly, MGMT methylated tumors are more diffuse than unmethylated tumors (p = 0.011). Among only unmethylated tumors, we again see the pattern that responders tend to have tumors that are more nodular.

Since our comparison of MGMT status and pre-adjuvant D/rho had a relatively small sample size, we identified 49 additional first-diagnosis GBM patients (who were excluded from other analyses because they did not meet the post-adjuvant imaging inclusion criteria) with available MGMT status and pre-adjuvant D/rho from our database for validation. In this combined cohort (23 patients who met inclusion criteria plus the 49 additional patients for this particular analysis), MGMT methylated patients (n = 28) had tumors that were significantly more diffuse than those in unmethylated patients (n = 44) (p = 0.006) (S6 Fig). This

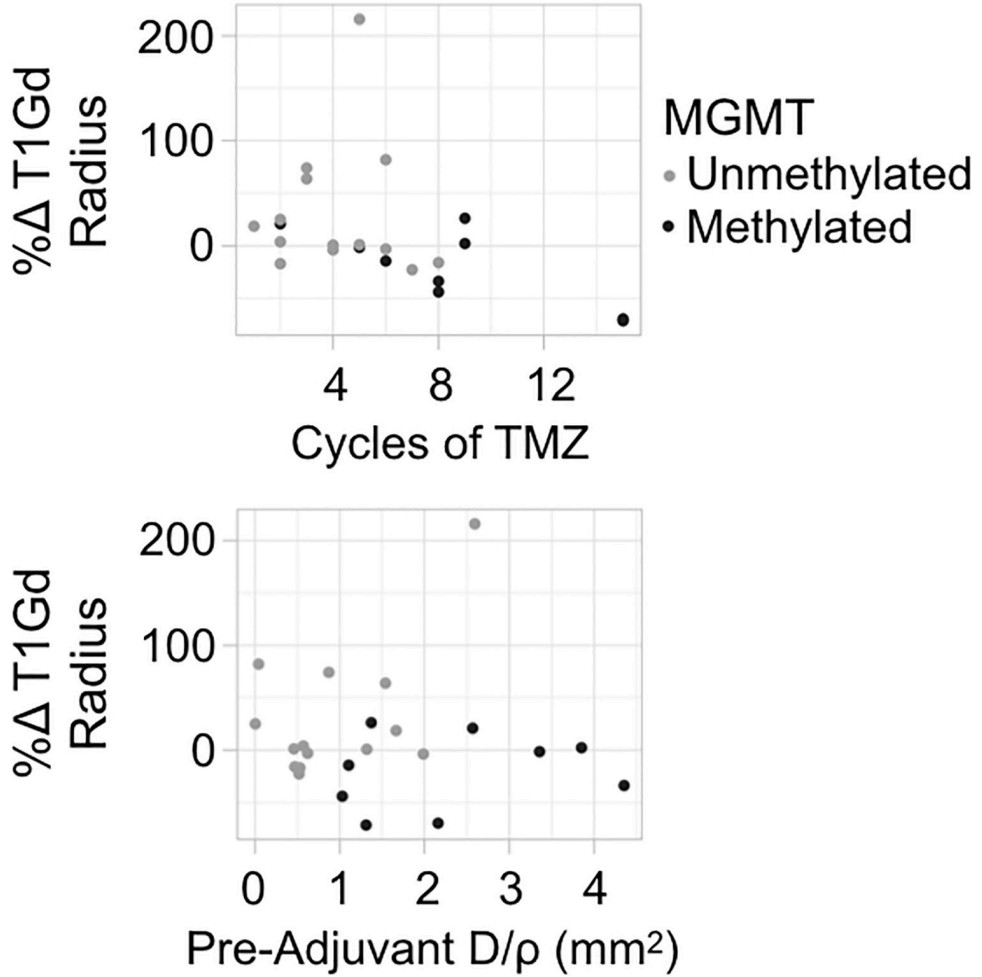

**Fig 5. Percent change T1Gd radius vs cycles of TMZ and pre-adjuvant D/rho by MGMT methylation status.**
Methylated patients (n = 9) have a clear negative trend between cycles of TMZ and volumetric response, while
unmethylated patients (n = 14) show a similar trend, but with more deviance. Methylated patients have more diffuse
tumors (higher pre-adjuvant D/rho) than unmethylated. Note that these trends are only observations, as this
comparison lacks statistical power due to small sample size.

confirms that the relationship between nodularity and response is not confounded by a pre-
dominance of methylated tumors in the nodular group.

## Using volume change until nadir validates previous results

We investigated whether analyzing the nadir (lowest T1Gd volume during adjuvant therapy)
time point instead of the post-adjuvant time point would be more informative (S2 Fig). Using
volume change between the pre-adjuvant T1Gd image and the nadir image for calculating the
percent change in T1Gd radius (%Δ T1Gd-Nadir), we found the similar results to those shown
in Fig 4. Specifically, among nodular tumors there was a significant negative correlation between
number of TMZ cycles received and %Δ T1Gd-Nadir (F-test, p<0.0001), and this relationship
was not significant among diffuse tumors (p = 0.1610) (S3 Fig). We also found that the overall
volumetric change (from the pre-adjuvant to post-adjuvant time points) was more closely tied to
clinical outcome than the change from pre-adjuvant imaging to nadir.

In order to assess the potential impact of pseudoprogression on our results, we performed all of the above analyses on a subcohort of patients that had at least 12 weeks between the XRT end date and the date of post-adjuvant imaging (n = 72) (S7–S10 Figs). This investigation showed comparable results with the full cohort, with the sole exception being the comparison of pre-adjuvant D/rho between responders and non-responders, which only trended towards significance (p = 0.0658).

## Discussion

In this investigation, we examined a number of patient attributes to assess whether any might be predictive of response to adjuvant TMZ. We found that patients whose T1Gd abnormality decreased in volume during adjuvant TMZ therapy were younger in age, received more cycles of TMZ, had longer time to nadir, and had more nodular tumors than those whose abnormality increased in volume. This decrease in volume was associated with better outcomes, including longer overall survival and a trend towards longer time to progression compared to those that had an increase in volume.

Some of these findings were expected and consistent with earlier studies. For example, younger patients have been shown to have better outcomes in other studies [44]. While this could be caused by differences in chemotherapy tolerance, a more favorable volumetric response to chemotherapy could also contribute to the survival differences observed between older and younger GBM patients. Additionally, while toxicity remains a concern, our finding that increased cycles of TMZ correlates with volumetric response supports other studies showing that more cycles of TMZ result in better response and outcomes [45–47].

Other findings were less intuitive, but also consistent with earlier studies. The association of longer time to nadir with response to TMZ, while not expected, is consistent with longer durability of TMZ effect upon tumor. Initial pseudoprogression may also contribute to the observation that volumetric responders reached nadir volume later in their adjuvant cycling. Our analyses that used volume change between pre-adjuvant and nadir images supported the results from the analyses that used the change between pre-adjuvant and post-adjuvant volumes.

The most important and unanticipated finding of this work was that nodular tumors tend to respond more favorably to adjuvant TMZ, both in terms of volumetric change and outcomes. Patients whose T1Gd abnormality decreased in size over the course of adjuvant therapy had significantly more nodular tumors than those who had an increase in size. Furthermore, patients with more nodular tumors had a clear negative correlation between cycles of TMZ received and volumetric response, with more cycles of TMZ resulting in a more favorable volumetric response. Among diffuse tumors, this relationship was neither visibly clear nor statistically significant. When looking at clinical outcomes, volumetric responders had significantly longer overall survival compared to volumetric non-responders among patients with nodular tumors, while this comparison was not significant among patients with diffuse tumors.

It has been previously suggested that TMZ might be less effective in more diffuse tumors. This could be due to reduced proliferation relative to invasion or to reduced drug distribution in these tumors. At least one study has suggested that TMZ might be present in higher concentrations near the contrast-enhancing core of the tumor, where the BBB is more likely to be compromised, compared to the surrounding tissue [8,48]. In a nodular tumor, a larger proportion of the visible tumor cells (on MR imaging) are near the contrast–enhancing core of the tumor, while in diffuse tumors, there are thought to be more image–detected invasive cells in the periphery [49], potentially limiting the efficacy of TMZ. We suspect this is the most likely explanation for the observations we made in this investigation, particularly since our invasion metric is a relative measure of invasion to proliferation and not proliferation outright;

however, more research is needed to fully understand how tumor characteristics interact with the BBB to affect drug distribution in brain tissue.

Lack of MGMT expression is mechanistically linked to TMZ sensitivity, and MGMT promoter methylation results in more favorable responses to TMZ chemotherapy [50–54]. Although limited by the small sample size of our patient cohort with known MGMT status, we wanted to ensure that the relationship between nodularity and responsiveness to TMZ was not confounded by MGMT methylation. We found that the MGMT methylated tumors were more diffuse at the pre-adjuvant imaging time point than the unmethylated tumors (S5 Fig). When we expanded our cohort to include more than seventy patients with MGMT status, we found that this observation remained true (S6 Fig). Therefore, we concluded that since the presumably more responsive MGMT methylated tumors were concentrated in the diffuse group, the observation that nodular tumors respond better to TMZ is not likely confounded by this molecular marker.

### Limitations and future work

We acknowledge that our retrospective investigation has some limitations and hope that after independent replication, these results will be validated and become clinically applicable. Further, our response classification (responder vs. non-responder) is not intended for clinical use and is not meant to distinguish progression or stable disease or to be used for clinical decision–making. We focused on quantifiable imageable response to the exclusion of nuanced clinical aspects of patient response, such as performance status and steroid use, that are needed in clinical metrics like RANO. Despite the simplicity of our metric and the potential for pseudoprogression to confound its results, it remained closely tied to outcome, which we believe justifies its use in a retrospective analysis. While some patients did receive other therapies during their adjuvant TMZ, this only occurred in a 12.2% of cases and usually towards the end of adjuvant therapy. Further, no patients received other therapies during the analyzed imaging periods (as noted in the Methods, for those 11 subjects, we restricted our analysis to the period during which no other anti-tumor therapies were administered).

Future work could attempt to identify other tumor characteristics that correspond to TMZ response. Notably, our investigation did not find a relationship between changes in the T2-FLAIR abnormality during adjuvant TMZ and tumor characteristics or patient outcome. While T2-FLAIR identifies fluid, this could be associated with extracellular fluid from leaky vasculature, immune recruitment and inflammation, or perhaps some other process. Each of these has different biological implications and more research is needed to uncover T2-FLAIR image features that indicate which of these processes are being visualized and to explore the different clinical implications of these processes. Future work could also look for sex differences in TMZ response. The results of previous work on sex differences suggests that TMZ might have sex-specific effects [55,56], which we hypothesized might affect the tumors in this cohort. When we ran the tests from this investigation on male and female patients separately, we observed that the same trends remained significant in the male cohort and were mostly insignificant in the female cohort (S11–S14 Figs). However, the small size of our female sample limits our ability to draw conclusions from this observation.

### Conclusion

In our retrospective investigation, we found that factors like patient age, cycles of TMZ received, time to nadir volume, and tumor nodularity are associated with volumetric response during adjuvant TMZ in GBM patients receiving standard of care treatment. Most notably, we found that nodular tumors have a cycle-dependent and more favorable image-based response

to TMZ compared to diffuse tumors. While MGMT methylation is often considered to predict a positive response to TMZ, our results suggest that nodularity may also serve as a predictor of response, especially among unmethylated tumors.

## Supporting information

**S1 Fig. Pre- and post-adjuvant imaging time points in relation to treatment time points.** Top row: In the case of no other treatment until later recurrence, the post-adjuvant image is the first image collected following the last cycle of TMZ. Middle row: in the case of co-administered therapy, the post-adjuvant image is the first image collected following the last cycle of TMZ administered alone/before an additional therapy was introduced. Last row: any cycles of TMZ given after an intervening time on another therapy was excluded from our analysis. (DOCX)

**S2 Fig. Schematic showing T1Gd radius changes from the pre-adjuvant to post-adjuvant imaging time points to demonstrate how nadir is determined and how proportion of adjuvant to nadir is calculated.** Proportion of adjuvant to nadir is calculated as number of days from pre-adjuvant image to nadir image divided by number of days between pre-adjuvant and post-adjuvant imaging. Top row: Example of tumor reaching nadir volume before post-adjuvant imaging, where %ΔT1Gd-Nadir (about -0.95) is less than %ΔT1Gd (about -0.50). Middle: Example of tumor reaching nadir at post-adjuvant time point, where %ΔT1Gd-Nadir equals %ΔT1Gd. Bottom: Example of how nadir is determined when tumor volume never decreases below pre-adjuvant value. (DOCX)

**S3 Fig. Percent change in T1Gd signal until nadir vs cycles of TMZ.** The nodular tumors have a negative correlation between number of TMZ cycles received and percent change of T1Gd radius from pre-adjuvant imaging to nadir imaging (F-test, $p < 0.0001$), while the diffuse tumors do not have a significant trend ($p = 0.161$), supporting the results from Fig 4. (DOCX)

**S4 Fig. MGMT methylation status.** Although the small sample size limits the statistical power of this comparison, methylated responders tended to have the best survival outcomes compared to other groups, followed by methylated non-responders, unmethylated responders, and unmethylated non-responders. (DOCX)

**S5 Fig. Methylated tumors are more diffuse than unmethylated tumors.** Among unmethylated tumors, four of the five responders are quite nodular, while the non-responders are distributed more evenly. Methylated patients received significantly more cycles of TMZ than unmethylated patients (t-test $p = 0.0156$). Among unmethylated patients, responders tended to receive more cycles of TMZ than non-responders. (DOCX)

**S6 Fig. Distribution of pre-adjuvant D/rho vs MGMT methylation status in an expanded cohort.** This expanded cohort (44 unmethylated, 28 methylated) uses a larger sample size to validate the observation that MGMT methylated tumors are significantly more diffuse than MGMT unmethylated tumors at the pre-adjuvant time point. Therefore, we do not think that the relationship between nodularity and response to TMZ is confounded by a predominance of MGMT methylated tumors in the nodular groups. Patients in this cohort are the 23 MGMT patients from the original patient cohort plus 49 additional patients with available MGMT status and pre-adjuvant D/rho from our database. These additional patients (30 unmethylated

and 19 methylated) were first diagnosis GBM patients who received maximal safe resection followed by concurrent radiotherapy (XRT) and chemotherapy (TMZ). Pre-adjuvant d/rho was calculated from the T1Gd and T2-FLAIR images taken after concurrent therapy ended and before adjuvant therapy began. Patients who received therapies other than XRT, TMZ, steroids, or anti-seizure medications before the image was taken were not included in this cohort.
(DOCX)

**S7 Fig. Characteristic differences between responders (n = 38) and non-responders (n = 34) for subjects with more than 12 weeks between end of XRT and post-adjuvant imaging.** These results are almost the same as those in Fig 1, except the pre-adjuvant D/rho comparison is no longer statistically significant (p = 0.066).
(DOCX)

**S8 Fig. Overall survival and progression-free survival for subcohort with more than 12 weeks between end of XRT and post-adjuvant imaging (n = 72).** Outcome differences between responders (n = 38) and non-responders (n = 34), show similar results as those in Fig 3.
(DOCX)

**S9 Fig. Percent change T1Gd radius and cycles of TMZ as well as survival probabilities for responders and non-responders grouped as nodular, moderate, and diffuse pre-adjuvant tumors for subjects with more than 12 weeks between end of XRT and post-adjuvant imaging (n = 72).** Similar to the results of Fig 4, nodular tumors (n = 24) show a significant negative correlation between cycles of TMZ received and change in tumor size and this change in tumor size results in a significant survival benefit. Neither the trend nor the survival benefit are observed among diffuse tumors (n = 24).
(DOCX)

**S10 Fig. Percent change T1Gd radius vs cycles of TMZ and pre-adjuvant D/rho by MGMT methylation status for subjects with more than 12 weeks between end of XRT and post-adjuvant imaging and methylation status available (n = 19).** Similar to Fig 5, methylated patients (n = 9) show a clearer correlation between cycles of TMZ received and change in tumor size. Unmethylated tumors (n = 10). tend to be more nodular compared to the methylated ones.
(DOCX)

**S11 Fig. Fig 1 split into males and females.** Comparison of characteristics between responders (male n = 28, female n = 17) and non-responders (male n = 32, female n = 13). Males have the same results as the combined population, while the female tests were insignificant.
(DOCX)

**S12 Fig. Fig 3 split into males and females.** Male responders (n = 28) had better overall survival than male non-responders (n = 32), while females did not have a significant survival difference between responders (n = 17) and non-responders (n = 13). Neither males nor females had a significant difference in progression free survival between responders (male n = 15, female n = 1) and non-responders (male n = 17, female n = 2).
(DOCX)

**S13 Fig. Fig 4 split into males and females.** Similar to the combined population, males with nodular tumors (n = 20) have a significantly negative trend between cycles of TMZ and volumetric change and this volumetric change results in a survival difference, while neither of these observations are significant among diffuse male tumors (n = 20). Visually, females show similar trends as the combined population, but the small female population size (nodular

n = 10, diffuse n = 10) likely contributes to the statistical insignificance of these observations.
(DOCX)

**S14 Fig. Fig 5 split into males and females.** Males (unmethylated n = 12, methylated n = 7) show similar trends between methylation status, volumetric change, diffusivity, and cycles of TMZ as the larger population, while there are not enough females with methylation status available (unmethylated n = 2, methylated n = 2) to draw a conclusion.
(DOCX)

**S1 Table. Distributions and counts of variables related to nadir analysis.** Nadir is defined as the lowest volume on T1Gd imaging (converted to spherically equivalent radii for analysis) after the pre-adjuvant date and on or before the post-adjuvant date. [b]For example, a value of 1 means that nadir occurred on the post-adjuvant image date, 100% of the way through adjuvant therapy. A value of 0.5, means that nadir occurred halfway through adjuvant therapy.
(DOCX)

**S1 Dataset. Minimal de-identified dataset.** Includes sheet with the 90 subjects meeting our original inclusion criteria and another sheet with additional subjects for the supplemental MGMT investigation. Key to coded fields: resection. grade: 3 = GTR, 2 = STR, 1 = biopsy; MGMT. Status: NA = not available, 0 = undetermined, 1 = unmethylated, 2 = methylated; X6monthgap.11pts: NA = no gap between cycles of TMZ therapy, 1 = gap between some TMZ cycles of 6 months or more.
(XLSX)

## Author Contributions

**Conceptualization:** Susan Christine Massey, Haylye White, Paula Whitmire, Kristin R. Swanson.

**Data curation:** Sandra K. Johnston, Pamela R. Jackson, Bernard R. Bendok, Alyx B. Porter, Sujay Vora, Jann N. Sarkaria, Leland S. Hu, Maciej M. Mrugala, Kristin R. Swanson.

**Formal analysis:** Susan Christine Massey, Paula Whitmire, Kristin R. Swanson.

**Funding acquisition:** Jann N. Sarkaria, Leland S. Hu, Kristin R. Swanson.

**Investigation:** Susan Christine Massey, Haylye White, Paula Whitmire, Tatum Doyle, Sandra K. Johnston, Kyle W. Singleton, Andrea Hawkins-Daarud, Bernard R. Bendok, Alyx B. Porter, Jann N. Sarkaria, Leland S. Hu, Maciej M. Mrugala.

**Methodology:** Susan Christine Massey, Haylye White, Paula Whitmire, Kristin R. Swanson.

**Resources:** Pamela R. Jackson, Leland S. Hu, Kristin R. Swanson.

**Software:** Haylye White, Kyle W. Singleton, Andrea Hawkins-Daarud.

**Visualization:** Paula Whitmire.

**Writing – original draft:** Susan Christine Massey, Paula Whitmire, Tatum Doyle.

**Writing – review & editing:** Haylye White, Sandra K. Johnston, Kyle W. Singleton, Pamela R. Jackson, Andrea Hawkins-Daarud, Bernard R. Bendok, Alyx B. Porter, Sujay Vora, Jann N. Sarkaria, Leland S. Hu, Maciej M. Mrugala, Kristin R. Swanson.

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
