## [Decision Letter · Decision Letter 0]

14 Oct 2019

PONE-D-19-22273

Image-based metric of invasiveness predicts response to adjuvant temozolomide for primary glioblastoma

PLOS ONE

Dear Dr. Massey,

Thank you for submitting your manuscript to PLOS ONE. After careful consideration, we feel that it has merit but does not fully meet PLOS ONE’s publication criteria as it currently stands. Therefore, we invite you to submit a revised version of the manuscript that addresses the points raised during the review process.

Please discuss the objections and suggestions for improvement brought forward by the reviewers. Please amend the manuscript accordingly where possible.

We would appreciate receiving your revised manuscript by Nov 28 2019 11:59PM. To enhance the reproducibility of your results, we recommend that if applicable you deposit your laboratory protocols in protocols.io, where a protocol can be assigned its own identifier (DOI) such that it can be cited independently in the future. For instructions see: http://journals.plos.org/plosone/s/submission-guidelines#loc-laboratory-protocols

We look forward to receiving your revised manuscript.

Kind regards,

Michael C Burger, M.D.

Academic Editor

PLOS ONE

Journal Requirements:

3. Please provide additional details regarding participant consent. In the ethics statement in the Methods and online submission information, please ensure that you have specified (1) whether consent was informed and (2) what type you obtained (for instance, written or verbal, and if verbal, how it was documented and witnessed). If your study included minors, state whether you obtained consent from parents or guardians. If the need for consent was waived, please ensure that you have discussed whether all data were fully anonymized before you accessed them and/or whether the IRB or ethics committee waived the requirement for informed consent.

Reviewers' comments:

Reviewer's Responses to Questions

**Comments to the Author**

1. Is the manuscript technically sound, and do the data support the conclusions?

Reviewer #1: Yes

Reviewer #2: Partly

2. Has the statistical analysis been performed appropriately and rigorously? 

Reviewer #1: I Don't Know

Reviewer #2: I Don't Know

3. Have the authors made all data underlying the findings in their manuscript fully available?

Reviewer #1: No

Reviewer #2: No

4. Is the manuscript presented in an intelligible fashion and written in standard English?

Reviewer #1: Yes

Reviewer #2: Yes

5. Review Comments to the Author

Reviewer #1: Using volumetric changes in MR imaging abnormalities during adjuvant therapy with TMZ Massey et al. study in their articel the relationships between several patient and tumor characteristics as well as volumetric and survival outcomes. Despite the long-term use of temozolomide in glioblastoma therapy, the optimal use of TMZ is still unclear. The authors report that factors like patient age, cyclyes of TMZ received, time to nadir volume and tumor nodularity are associated with volumetric response during adjuvant TMZ therapy in glioblastoma patients. Some of these aspects are already known. The main novel aspect of this article is that nodular tumors have a cycle-dependent and more favorable image-based response to TMZ than diffuse tumors. The main disadvantage of this study is that the division into responder vs. non-responder only refers to Gd-enhancement and thus can not be used in everyday clinical practice.

While reading some questions arise:

Please describe the tumor segmentation process more exactly. Which software was used for this? Have all patients been segmented by one person? If not, how were intersegmenter differences resolved?

In the results section under "T1Gd volumetric response correlates with younger patient age ...", please specify the value and mean and median of the results of the two different groups if applicable as a table. Since at present only the p-value and a digram are shown, but a comparison of the values per se is not possible.

Where was the cut off of the D / rho value for the 3 different groups: "nodular, moderate and diffuse" set?

Reviewer #2: Page 4

„overall survival of 15 months [3,4].“

Maybe a more recent study would be suitable, e.g. control group of EF-14 data which exceeds the 15 months.

„The similar radiological presentation of growing tumor and pseudoprogression complicates the assessment of TMZ response [9]. „

Please comment on the role of MGMT promotor methylation status (more frequent pseudprogression, role of PET to differentiate tumor from pseudoprogression etc.)

„each 28–day adjuvant cycle for 6–12 cycles.“

Please specify: 6 cycles in Stupp protocol, 12 cycles in CATNON protocol.

Page 5

„although the number of cycles of adjuvant TMZ administered in clinical practice varies.“

However, recent meta-analysis did not show superiority of more than 6 compared to only 6 cyles TMZ.

„found that adding adjuvant TMZ to a radiotherapy treatment course improves median overall survival by 3.7 months and PFS by 5.4 months [24]. „

Maybe a quote of the Perry trial (40 Gy in 15 fractions of radiotherapy plus 6 adjuvant cycles of TMZ) or Nordic trial (10x 3.4 Gy vs. TMZ alone) would add to treatment in the elderly.

Temozolomide versus standard 6-week radiotherapy versus hypofractionated radiotherapy in patients older than 60 years with glioblastoma: the Nordic randomised, phase 3 trial.

Malmström A, Grønberg BH, Marosi C, Stupp R, Frappaz D, Schultz H, Abacioglu U, Tavelin B, Lhermitte B, Hegi ME, Rosell J, Henriksson R; Nordic Clinical Brain Tumour Study Group (NCBTSG).

Lancet Oncol. 2012 Sep;13(9):916-26. Epub 2012 Aug 8.

Short-Course Radiation plus Temozolomide in Elderly Patients with Glioblastoma.

Perry JR, Laperriere N, O'Callaghan CJ, Brandes AA, Menten J, Phillips C, Fay M, Nishikawa R, Cairncross JG, Roa W, Osoba D, Rossiter JP, Sahgal A, Hirte H, Laigle-Donadey F, Franceschi E, Chinot O, Golfinopoulos V, Fariselli L, Wick A, Feuvret L, Back M, Tills M, Winch C, Baumert BG, Wick W, Ding K, Mason WP; Trial Investigators.

N Engl J Med. 2017 Mar 16;376(11):1027-1037. doi: 10.1056/NEJMoa1611977.

„Eleven patients received a therapy other than TMZ concurrent with or in between cycles of TMZ“

These patients should be excluded from analysis.

What was the minimum / maximum and mean number of TMZ cycles before radiographic response analysis? How did you check that „late“ response during adjuvant therapy was not missed in these 11?

Table 1

Half of the patients received less than 6 cycles of TMZ and only half oft he patients had progression(!). Numbers are small…

Was progression confirmed by histology or clinical follow-up?

„Extent of resection is abstracted from surgical notes and radiological reports and is not uniformly determined radiographically.“

EOR should be calculated by volumetrics and NOT by surgeon‘s notes.

Page 8/9

In GTR no residual Gad enhancement should be notable. How did you calculate „no residual Gad enhancement“? Was early postoperative MRI used in all cases? Or was there another scan before starting radiotherapy?

RANO criteria allow 25% volume increase compared to baseline to call it stable disease. These would be non-responders in your calculation. Please comment.

„While this remains a potential confounder for many GBM studies, our classification correlated well with overall and progression free survival, suggesting that any such misclassification was minimal.“

How do you know that? Biopsy? PET? Regional CBF/CBV measurements?

Results

Fig 2: Did you use subtraction images (T1 post Gad minus T1 without Gad) to rule out hemorrhage in the baseline scan as „tumor“?

The nodular responder would be called progressive disease using RANO (FLAIR lesion plus >25% increase in size).

Fig 3:

Why did half oft the patient progress within 3 months? This is very early.

Page 14:

Which method was used for MGMT promotor methylation? What was your cut-off to call it methylated? Were all tumors IDH-1 wildtype? Any other molecular data like TERT? How do you know there was no IDH-mutated glioma included?

Page 19:

„Therefore, we concluded that since the presumably more responsive MGMT methylated tumors were concentrated in the diffuse group, the observation that nodular tumors respond better to TMZ is not likely confounded by this molecular marker.“

But this would mean the nodular tumors are rather MGMT-promotor non-methylated and should respond WORSE to TMZ? Does that make sense?

Page 20:

„While some patients did receive other therapies during their adjuvant TMZ, this only occurred in a small number of cases.“

11/90 is not a small number.

„While T2-FLAIR identifies fluid, this could be associated with extracellular fluid from leaky vasculature, immune recruitment and inflammation, or perhaps some other process“

…namely non-enhancing tumor.

I think your sample size is too heterogeneous and uncharacterized (IDH, MGMT, resection status, MRI data) to draw any conclusions. Missing data should be added (e.g. # of TMZ cycles). Then your conclusions should be validatetd in another cohort.

6. PLOS authors have the option to publish the peer review history of their article (what does this mean?). If published, this will include your full peer review and any attached files.

Reviewer #1: No

Reviewer #2: Yes: Martin Misch

---

## [Author Response · Author response to Decision Letter 0]

25 Feb 2020

Reviewer #1: Using volumetric changes in MR imaging abnormalities during adjuvant therapy with TMZ Massey et al. study in their articel the relationships between several patient and tumor characteristics as well as volumetric and survival outcomes. Despite the long-term use of temozolomide in glioblastoma therapy, the optimal use of TMZ is still unclear. The authors report that factors like patient age, cyclyes of TMZ received, time to nadir volume and tumor nodularity are associated with volumetric response during adjuvant TMZ therapy in glioblastoma patients. Some of these aspects are already known. The main novel aspect of this article is that nodular tumors have a cycle-dependent and more favorable image-based response to TMZ than diffuse tumors. The main disadvantage of this study is that the division into responder vs. non-responder only refers to Gd-enhancement and thus can not be used in everyday clinical practice.

We thank the reviewer for their thoughtful response to our paper. We would like to note that although our metric cannot be used in the way that clinical evaluation criteria are used (which incorporates other factors such as patient symptoms), we believe that this information may be helpful to clinicians who are considering whether a patient may benefit from receiving more cycles of TMZ (particularly for patients whose tumors are MGMT unmethylated).

While reading some questions arise:

Please describe the tumor segmentation process more exactly. Which software was used for this? Have all patients been segmented by one person? If not, how were intersegmenter differences resolved?

We thank the reviewer for noting that this was not sufficiently well described. We have revised the text (in the Methods/Imaging and Biomathematical Model section, page 8) to better describe the segmentation process, particularly addressing quality assurance procedures.

In the results section under "T1Gd volumetric response correlates with younger patient age ...", please specify the value and mean and median of the results of the two different groups if applicable as a table. Since at present only the p-value and a digram are shown, but a comparison of the values per se is not possible.

We thank the reader for noting that we perhaps did not describe the violin plots of Fig 1 sufficiently well. We have added notes in the legend that the small black dots correspond to individual subjects, the diamond indicates the median value, and the outline denotes the frequency (wider regions have higher frequency than narrow regions). While the resolution of the axes may not lead to precise calculation of the mean and median, these plots show that although the responder group is younger in a statistically significant manner, the means between the two are less than 10 years apart, and the age distributions span similar ranges. We felt this was more informative than reporting values in a table. 

Where was the cut off of the D / rho value for the 3 different groups: "nodular, moderate and diffuse" set?

As described in the Fig 4 legend and the preceding text (Results, sub-section “Increasing number of cycles correlates with volumetric response only in nodular tumors”), the cut offs were not pre-selected, but we instead divided subjects into three equally sized groups. Thus, the lowest third of D/rho values comprised the “nodular” group, the highest third comprised the “diffuse” group, and the middle third was the “moderate” group. We chose to do this (instead of focusing on particular threshold values), as we felt this would be more clinically translatable for settings where segmentation and quantification of a precise D/rho value may not be practical, but a neuro-radiologist may be able to note that a tumor leans toward a more nodular or more diffuse pattern when comparing T1Gd and T2/FLAIR images. That said, to improve reproducibility among research groups who may also wish to compute this metric, we have added the D/rho value groupings in the legend of Fig 4 (most nodular group was 0.0034 to 0.572 mm2, moderate group was 0.6195 to 2.562 mm2, and most diffuse group was 2.567 to 9.53 mm2). 

Reviewer #2: 

Page 4: “overall survival of 15 months [3,4].“ Maybe a more recent study would be suitable, e.g. control group of EF-14 data which exceeds the 15 months. 

We thank the reviewer for this suggestion. We have revised this statement, reporting the survival statistics from reference [2], which is the most recent report issued for 5-year survival statistics in the United States (2012-2016, recently updated from the one used when this manuscript was originally submitted that analyzed years 2011-2015). Additionally, we have reported the survival times of the “Stupp protocol” group from both Stupp et al. 2005 and the EF-14 trial (Stupp et al. 2017). 

“The similar radiological presentation of growing tumor and pseudoprogression complicates the assessment of TMZ response [9].” Please comment on the role of MGMT promotor methylation status (more frequent pseudprogression, role of PET to differentiate tumor from pseudoprogression etc.)

We agree that incorporating these would be very interesting in a future prospective analysis. However, this is beyond the scope of our present study and we are concerned that adding this to the introduction of our manuscript may be misleading to readers (since, for example we did not have PET imaging available).

“each 28–day adjuvant cycle for 6–12 cycles.” Please specify: 6 cycles in Stupp protocol, 12 cycles in CATNON protocol.

To be clear, our intent was to reference actual TMZ cycles administered, which do not necessarily adhere to either of these particular cycle counts (due to factors such as side effects), but typically fall within this range. That is, among our neuro-oncology colleagues at a number of institutions, it is common practice to give TMZ for more than 6 cycles if it is well-tolerated and the clinician determines it may continue to benefit the patient. Additionally, the CATNON protocol is focused on anaplastic astrocytoma without 1p/19q co-deletion, and our analysis was only for patients diagnosed with glioblastoma, so we have chosen not to reference this study. 

Page 5 “although the number of cycles of adjuvant TMZ administered in clinical practice varies.” However, recent meta-analysis did not show superiority of more than 6 compared to only 6 cyles TMZ.

The reviewer does not indicate to which study they are referring. However, as we noted in the sentences following this selected excerpt, some studies and meta-analyses have shown benefit while other studies have not. Taken together, these might suggest that some patients may benefit more than others (and determining which patients might benefit from additional TMZ, particularly among those without MGMT methylation, was the primary objective for our present investigation). We have added a statement regarding this latter point as a final sentence to the referenced paragraph (splits across pages 4-5). 

“found that adding adjuvant TMZ to a radiotherapy treatment course improves median overall survival by 3.7 months and PFS by 5.4 months [24].” Maybe a quote of the Perry trial (40 Gy in 15 fractions of radiotherapy plus 6 adjuvant cycles of TMZ) or Nordic trial (10x 3.4 Gy vs. TMZ alone) would add to treatment in the elderly.

Temozolomide versus standard 6-week radiotherapy versus hypofractionated radiotherapy in patients older than 60 years with glioblastoma: the Nordic randomised, phase 3 trial. Malmström A, Grønberg BH, Marosi C, Stupp R, Frappaz D, Schultz H, Abacioglu U, Tavelin B, Lhermitte B, Hegi ME, Rosell J, Henriksson R; Nordic Clinical Brain Tumour Study Group (NCBTSG). Lancet Oncol. 2012 Sep;13(9):916-26. Epub 2012 Aug 8.

Short-Course Radiation plus Temozolomide in Elderly Patients with Glioblastoma. Perry JR, Laperriere N, O'Callaghan CJ, Brandes AA, Menten J, Phillips C, Fay M, Nishikawa R, Cairncross JG, Roa W, Osoba D, Rossiter JP, Sahgal A, Hirte H, Laigle-Donadey F, Franceschi E, Chinot O, Golfinopoulos V, Fariselli L, Wick A, Feuvret L, Back M, Tills M, Winch C, Baumert BG, Wick W, Ding K, Mason WP; Trial Investigators. N Engl J Med. 2017 Mar 16;376(11):1027-1037. doi: 10.1056/NEJMoa1611977.

Thank you for pointing out these additional studies investigating TMZ in elderly patients. While these studies primarily focus on alternative radiotherapy regimens, they do also further support the use of TMZ in treating older patients. The references have been added (page 6).

“Eleven patients received a therapy other than TMZ concurrent with or in between cycles of TMZ” These patients should be excluded from analysis.

We had extensive discussion on this point amongst ourselves (the co-authors) prior to deciding on including these individuals and based on that, we respectfully disagree with the reviewer in this case. We decided to include these subjects but simply evaluate the period during which they received only adjuvant TMZ in order to examine the percent change in T1Gd during this period, as we explained in the text (section Methods, Patient Cohort, page 7): 

“Further, to ensure that we captured the effect of TMZ exclusively in these eleven cases, the image before the start of the other therapy was used as the post-adjuvant image, so that no patients received other therapies during the analyzed imaging period.”

The decision was based on two main factors: (1) most, if not all, subjects go on to receive additional therapy following the adjuvant TMZ period (i.e., when they recur), and (2) our primary focus is on the impact of cycles of TMZ upon the size of T1Gd abnormality during the adjuvant alone treatment period (including the TMZ only time for those eleven patients), and whether that change is prognostically significant.

What was the minimum / maximum and mean number of TMZ cycles before radiographic response analysis? How did you check that “late” response during adjuvant therapy was not missed in these 11?

We have reported the number of cycles received in Table 1 and in Figures 1, 4, and 5, as well as discussed this in the results. As described on page 8 in the Imaging and Biomathematical Model subsection of the Methods, we counted the number of cycles received by each patient during the adjuvant period, and for those 11 subjects we considered the adjuvant period (for the purposes of our analysis) to have ended when additional therapies were received. Further, we are not evaluating the effects of therapy at late time points (again as discussed in the Imaging and Biomathematical Model subsection of the Methods). For even greater clarification, we have added a Supplementary Figure S1 illustrating the time points considered as pre- and post- adjuvant in our analysis. 

Table 1 - Half of the patients received less than 6 cycles of TMZ and only half oft he patients had progression(!). Numbers are small… Was progression confirmed by histology or clinical follow-up?

Regarding the number of TMZ cycles, it appears that the reviewer missed the text in the table footnote on this point: “The cycles of adjuvant TMZ reported here exclude any cycles that were given in conjunction with other anti-tumor therapies since these were excluded from our analysis (see Methods). It should be noted that the majority of patients did receive at least 6 cycles of TMZ, even if they were not counted for the adjuvant period in our analysis.” 

Regarding progression, our number refers to the number of subjects for whom the date of progression was known/determined (and distinguished as “true progression” from pseudoprogression), and not, as the reviewer appears to assume, a count of those who progressed. Since this was not explicitly stated, we have added a table footnote to clarify this: “These are results for subjects with a known date of true progression only, which was slightly less than half of the total cohort (47.8%).” Additionally, progression was determined clinically, usually on the basis of radiographic evidence, as not all cases of progression result in a resection or biopsy. 

“Extent of resection is abstracted from surgical notes and radiological reports and is not uniformly determined radiographically.” EOR should be calculated by volumetrics and NOT by surgeon‘s notes.

We agree with the reviewer that extent of resection should be quantitatively determined to the fullest extent possible. This is of course dependent on the acquisition of post-operative imaging, particularly while T1 brightness enables visualization of blood (to do T1 subtraction). For the majority of GTR cases, this was possible and noted in radiology reports; however when this was not available (i.e, these images were not acquired), we made use of later radiological estimates combined with surgical and medical record notes. Further, since this was not the focus of our analysis, we did not make availability of such postoperative imaging an inclusion criterion. We have merely reported it here in Table 1 for the interest of readers in order to describe the cohort overall. We have edited the table caption: “Extent of resection is abstracted from surgical notes and radiological reports and is not uniformly verified radiographically (though most GTR cases were verified via imaging).” 

Page 8/9 - In GTR no residual Gad enhancement should be notable. How did you calculate „no residual Gad enhancement“? Was early postoperative MRI used in all cases? Or was there another scan before starting radiotherapy?

This is similar to the previous comment, and in addressing that we have also addressed this particular point; see above.

RANO criteria allow 25% volume increase compared to baseline to call it stable disease. These would be non-responders in your calculation. Please comment.

As we discuss at several points in the manuscript (see sections Methods, Response Indicator and Discussion, Limitations and Future Work, pages 9 and 20, respectively), our evaluation is entirely different from that of RANO. RANO is meant to be used as a clinical tool for evaluating disease status. Our focus is purely on whether the size of T1Gd enhancement increased or decreased during the period of adjuvant treatment, and as such whether the size of the tumor shrunk due to therapy (“responders”) or grew in spite of therapy (“non-responders”). Clinical assessment tools such as RANO incorporate other clinical factors such as patient symptoms and uses different thresholds of tumor volume for making determinations as to whether a patient ought to be removed from a therapy. This is well beyond the scope of our present effort, which should certainly not replace such clinical evaluation criteria. Our effort is only intended to provide additional insight into whether certain patients might be expected to receive further benefit from additional cycles of temozolomide.

“While this remains a potential confounder for many GBM studies, our classification correlated well with overall and progression free survival, suggesting that any such misclassification was minimal.” How do you know that? Biopsy? PET? Regional CBF/CBV measurements?

It is unclear what the reviewer is asking here - as we state in the quoted sentence, we see a good agreement between classification and both overall and progression free survival, and take this agreement as a suggestion that any misclassification due to pseudoprogression is minimal (but certainly not as a guarantee, since indeed we did not have access to biopsy or PET). 

Additionally, as we have stated in the following sentence: “To further reduce the possibility of misclassification due to pseudoprogression, we re-performed all of our analyses in a supplemental investigation using a subcohort of patients with more than 12 weeks between the end date of XRT and date of post-adjuvant imaging (n=72).”

Results

Fig 2: Did you use subtraction images (T1 post Gad minus T1 without Gad) to rule out hemorrhage in the baseline scan as “tumor”?

Because the pre-adjuvant image is also post- concurrent chemoradiation, any hemorrhaging due to surgery at this time point would likely no longer be bright on T1. Therefore, we did not do T1 subtraction for these images.

The nodular responder would be called progressive disease using RANO (FLAIR lesion plus >25% increase in size).

As discussed in regards to an earlier comment, our imaging-based assessment is not intended to be in any way analogous to RANO, and we have stated as much repeatedly in the manuscript (see sections Methods, Response Indicator and Discussion, Limitations and Future Work, pages 9 and 20, respectively).

Fig 3: Why did half oft the patient progress within 3 months? This is very early.

First, this is only progression for subjects with known dates of progression (as noted previously), which may be disproportionately known for those patients who progressed early. Beyond that, we choose not to speculate further; we are simply reporting the data that we have. 

Page 14: Which method was used for MGMT promotor methylation? What was your cut-off to call it methylated? Were all tumors IDH-1 wildtype? Any other molecular data like TERT? How do you know there was no IDH-mutated glioma included?

For MGMT promoter methylation, we relied on clinical pathology or research study reports from the various institutions where subjects were recruited (for those subjects who had testing). In our retrospective data set, such molecular features were not available for all of the subjects since these were less commonly tested for as part of clinical care when some of the (earlier) subjects were accrued to our data set and they may have only been tested for these if they also participated in a separate research study. 

Page 19: “Therefore, we concluded that since the presumably more responsive MGMT methylated tumors were concentrated in the diffuse group, the observation that nodular tumors respond better to TMZ is not likely confounded by this molecular marker.” But this would mean the nodular tumors are rather MGMT-promotor non-methylated and should respond WORSE to TMZ? Does that make sense?

While MGMT methylation is a primary indicator of positive TMZ response, we conclude that our results indicate tumor nodularity (vs diffusivity) may be a secondary indicator of positive TMZ response, particularly among MGMT unmethylated tumors. This quoted sentence is intended to highlight that we do not see evidence that this secondary indicator is “contaminated” by the primary indicator (that is, it is not driven by overrepresentation of MGMT methylated tumors in the nodular group). While MGMT methylation is indeed a clear prognostic indicator, we do not agree with the reviewer that all MGMT unmethylated tumors uniformly respond poorly to TMZ (if that is indeed the point being made); our evidence suggests that more nodular MGMT unmethylated tumors may benefit more from additional cycles of TMZ than diffuse unmethylated tumors.

Page 20: “While some patients did receive other therapies during their adjuvant TMZ, this only occurred in a small number of cases.“ 11/90 is not a small number.

That is indeed perhaps too subjective a descriptor; we have updated the statement with more precise, objective language: “...this only occurred in 12.2% of cases.” (page 20 in the edited manuscript)

“While T2-FLAIR identifies fluid, this could be associated with extracellular fluid from leaky vasculature, immune recruitment and inflammation, or perhaps some other process” …namely non-enhancing tumor.

We agree that T2-FLAIR is associated with non-enhancing tumor; this sentence is merely highlighting that there are various mechanisms within the tumor that could be contributing to this signal, which may vary across tumors. 

I think your sample size is too heterogeneous and uncharacterized (IDH, MGMT, resection status, MRI data) to draw any conclusions. Missing data should be added (e.g. # of TMZ cycles). Then your conclusions should be validatetd in another cohort.

We gladly welcome validation in another cohort, as we have clearly articulated in the Limitations and Future Work subsection of the Discussion (page 20). However, we are confused by the reviewer’s statement regarding missing data, since number of adjuvant TMZ cycles was in fact the data we have reported on for all of our subjects. While we concede that not all subjects’ tumors were molecularly characterized with MGMT status (due to historical clinical practice, see earlier comment on this point), we believe the cohort is quite consistent with respect to course of treatment and available pre- and post-adjuvant TMZ imaging, as detailed in the selection criteria in section Methods, Patient Cohort, pages 6-7.

---

## [Editor Report · Decision Letter 1]

3 Mar 2020

Image-based metric of invasiveness predicts response to adjuvant temozolomide for primary glioblastoma

PONE-D-19-22273R1

Dear Dr. Massey,

We are pleased to inform you that your manuscript has been judged scientifically suitable for publication and will be formally accepted for publication once it complies with all outstanding technical requirements.

With kind regards,

Michael C Burger, M.D.

Academic Editor

PLOS ONE
---

## [Editor Report · Acceptance letter]

13 Mar 2020

PONE-D-19-22273R1 

Image-based metric of invasiveness predicts response to adjuvant temozolomide for primary glioblastoma 

Dear Dr. Massey:

I am pleased to inform you that your manuscript has been deemed suitable for publication in PLOS ONE. Congratulations! Your manuscript is now with our production department. 

With kind regards,

on behalf of

Dr. Michael C Burger 

Academic Editor

PLOS ONE